# Nanomedicine Interventions in Clinical Trials for the Treatment of Metastatic Breast Cancer

Rita Moreira [1,2], Andreia Granja [3], Marina Pinheiro [3,*] and Salette Reis [3]

1 Faculdade de Medicina e Ciências Biomédicas, Universidade do Algarve, Campus de Gambelas, 8005-139 Faro, Portugal; ritamadalenapm@gmail.com
2 ABC-RI, Algarve Biomedical Center Research Institute, 8005-139 Faro, Portugal
3 LAQV, REQUIMTE, Departamento de Ciências Químicas, Faculdade de Farmácia, Rua de Jorge Viterbo Ferreira, 228, 4050-313 Porto, Portugal; andreia26293@gmail.com (A.G.); shreis@ff.up.pt (S.R.)
* Correspondence: marinabppinheiro@gmail.com

**Abstract:** Breast cancer was responsible for the deaths of 626,679 women in 2018. After decades of research, the mortality rates remain high. While the barrier of selectively killing tumor cells is not yet overcome, the search for targeted therapeutics continues. The use of nanomedicine in cancer treatment has opened up new possibilities for more precise drug-delivery systems. This review aimed to gather information and analyze recent clinical trials evaluating the therapeutic effects of nanoparticles in the treatment of metastatic breast cancer. To accomplish this, the clinicaltrials.gov database was consulted, and after employing specific exclusion criteria, 11 clinical trials were selected. Nanoparticle albumin-stabilized paclitaxel was evaluated in ten clinical trials and paclitaxel-incorporating polymeric micelles were assessed in one clinical trial. Overall, this review confirmed a clinical benefit in the use of nanoparticle albumin-stabilized paclitaxel for the treatment of breast cancer, with reduced toxicity when compared to first-line treatments. Three studies did not meet the primary endpoint, however, and so the authors advised further evaluations. Although the use of nanomedicine is revolutionizing the cancer field, to integrate this regimen into generalized clinical treatment, additional clinical trials must be performed to achieve a favorable safety and efficacy profile.

**Keywords:** breast cancer; clinical trials; drug delivery; metastasis; nanomedicine

## 1. Introduction

Cancer is one of the leading causes of death globally. Breast cancer is the most commonly diagnosed type of cancer in women (Figure 1) [1]. In the year 2018, 2.1 million women were diagnosed, and 626,679 women died due to this disease [2]. Its etiology is multifactorial, and some risk factors include age, family history, reproductive factors, use of certain contraceptives, and postmenopausal therapy. Despite the mortality rate remaining high, a declining trend has been observed for the last two decades due to early diagnosis, improved detection methods, and more sophisticated treatments [3]. Countries with premature implementation and high attendance of breast screenings have shown the highest decreases in mortality rate, despite the increasing incidence of cancers with low aggressive potential [4]. Still, a high mortality rate persists, possibly due the lack of therapeutic agents that act only on the target cells without damaging healthy cells [5].

Breast cancer is most commonly staged with the TMN system [2]. This system classifies the size of the primary tumor (T), the involvement of regional lymph nodes (N), and the presence or absence of metastasis (M) [6]. This classification creates five stages (0–IV) that describe the extent of the primary tumor and its metastasis [2]. Metastasis is a series of biological processes that start with the migration of tumor cells from the location of the primary tumor, followed by intravasation, survival, extravasation of the bloodstream, and

colonization of a distant site [7]. Metastatic breast cancer, despite being treatable, is a "technically" uncurable disease. In most patients with this condition, metastasis is the cause of death, with a median overall survival of 2–3 years [8]. Management of metastatic breast cancer differs according to its subtype [2]. There are more than 20 subtypes of breast cancer, according to the World Health Organization. The three major subtypes are hormone receptor-positive/HER2-negative (HR+/HER2−), HER2-positive (HER2+), and triple-negative [9]. Chemotherapy is generally used to prolong survival and to treat symptoms in patients with metastatic breast cancer. However, in monotherapy, it results in short survival and a brief response duration [10]. The management of these patients has evolved with the use of "precision" medicine, a technique that matches the treatment with the subtype of the tumor and patients' biomarkers [11]. For hormone receptor-positive/HER2-negative breast cancer, the initial therapeutical approach is endocrine treatment with or without targeted therapy. For all patients, first-line treatment consists of tamoxifen, an aromatase inhibitor, or fulvestrant. The addition of CDK 4/6 inhibitors improves PFS and overall survival [2]. Once endocrine resistance is achieved, a transition is made to chemotherapy with a single agent [9]. For HER2-positive, HER2-targeted agents, such as transtuzumab, and chemotherapy are first-line treatments [11]. In this subtype of breast cancer, brain metastasis is common and can be treated with both systemic and local therapy (radiation and surgery) [9]. Triple-negative breast cancer (TNBC) has one of the highest death rates, only second to HER2-positive breast cancer [2]. At a genetic level, TNBC is highly heterogeneous and tends to metastasize to the brain and lungs [12]. Since TNBC does not express estrogen receptors, progesterone receptors or HER2, "precision" therapy is not established. In this type of tumor, chemotherapy with platinum is the preferred option for treatment. For TNBC with >1% programmed death-ligand 1 (PD-L1) staining, abraxane with atezolizumab showed a high progression-free survival [2].

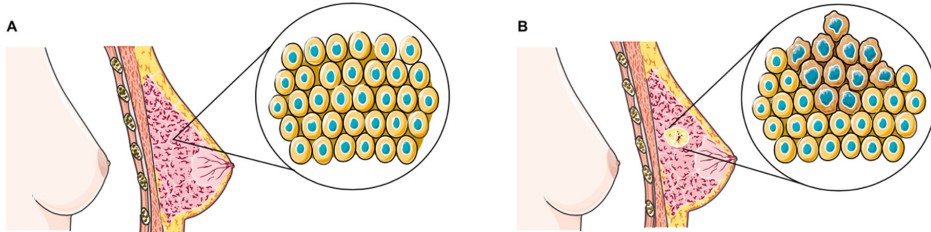

**Figure 1.** Representative image of a macroscopic and microscopic view of the breast and its tissue. (**A**) Healthy breast and tissue. (**B**) Breast with a tumor, only visible on a microscopic view. Images are modified from Servier Medical Art (https://smart.servier.com accessed on 20 December 2020).

Anticancer drugs are generally toxic to the healthy and tumor cells. Therefore, these treatments are restricted by their adverse effects (Table 1). The use of nanomedicine in cancer therapy has opened up new possibilities for more precise drug-delivery systems with fewer associated side-effects, and for more information on the impact of nanomedicine in the treatment of cancer, the reader can consult the following literature [5]. In particular, the authors highlight superparamagnetic iron oxide nanoparticles (SPIONs) that seem to have the potential for synergy with other methods of cancer therapy, including those discussed in this review [13–15]. However, since these technologies have not yet been introduced in clinical trials, they will not be discussed in this paper. Given the large amount of progress that has been made in the last few years, for the treatment of metastatic breast cancer using nanomedicine, this study sought to gather that information and compare it. Other studies have been published focusing on the use of nanocarriers for the treatment of breast cancer from a pre-clinical perspective [16]. This review aimed to synthesize the findings of the existing literature focusing on clinical trials, from the last 10 years, highlighting the prospective beneficial candidates in clinical practice in the treatment of metastatic breast cancer.

**Table 1.** Commonly used drugs in the treatment of breast cancer.

| Drug/Agent | Mechanisms | Half-Life | Route of Elimination | Common Side-Effects | Reference |
|---|---|---|---|---|---|
| Endocrine therapy | | | | | |
| Anastrazole | Non–steroidal aromatase inhibitor | 50 h | 85% in feces 10% in urine | Hot flashes, arthralgias or myalgias | [1,2] |
| Fulvestrant | Estrogen receptor antagonist | 40 days | ≈90% in feces <1% in urine | Alopecia, constipation, diarrhea, nausea and vomiting | [1,3] |
| Octreotide | Somatostatin analogue | 2.3–2.7 h | 32% in urine 30–40% in feces | Gastrointestinal, dizziness, dry skin and depressed mood | [1,3] |
| Tamoxifen | Selective estrogen receptor modulator | 5–7 days (14 days for its metabolite) | Mainly in feces | Hot flashes | [2] |
| Cytotoxic Chemotherapy | | | | | |
| Abraxane | Alkylating agente | 13–27 h | Mainly in bile | Hipersensitivity reaction, neutropenia, neuropathy and sepsis | [4] |
| Carboplatin | Alkylating agente | 1.1–5.9 h | Mainly in urine | Hypersensitivity reaction, nausea, vomiting, anemia and genitourinary symptoms | [1,3] |
| Cyclophosphamide | Alkylating agente | 3–12 h | Primarily in the form of metabolites 10–20% in urine unchanged | Neutropenia, febrile neutropenia, fever, alopecia, náusea, vomiting and diarrhea | [1] |
| Doxorubicin | Cytotoxic anthracycline antibiotic | 20–48 h | 40% in bile 5–12% in urine | Cardiomyopathy, myelosuppression, infection and septic shock | [1] |
| Gemcitabine | Anti–metabolite (nucleoside analog) | Short infusions: 42–94 min Long infusions: 245–638 min | 92–98% in urine | Alopecia, myelosuppression, nausea, vomiting and diarrhea | [1,3] |
| Paclitaxel | Antimicrotubule agent | 52.7 h (with a 24 h infusion) | ≈71% in feces ≈14% in urine | Bone marrow suppression, peripheral neurotoxicity and mucositis | [1] |
| Targeted Therapy | | | | | |
| Bevacizumab | Recombinant humanized IgG1 monoclonal antibody that against VEGF | ≈20 days | Information not found | Proteinuria, arterial thromboembolic events, GI bleeding and sepsis | [1] |
| Erlotinib | Inhibitor of EGFR tyrosine kinase | 36.2 h | 83% in feces 8% in urine | Diarrhea, rash and liver transaminase elevation | [1,3] |
| Irinotecan | Topoisomerase I inhibitor | 6–12 h | Bile and urine | Nausea, vomiting, abdominal cramping, diarrhea and infection | [1] |
| Lapatinib | 4–anilinoquinazoline kinase inhibitor of intracellular tyrosine kinase domains of HER1/EGFR/ERBB1 and HER2/ERBB2 | 14.2 h | 14% in feces | Diarrhea and vomiting | [1] |
| Trastuzumab | Recombinant humanized IgG1 monoclonal antibody against the HER2 receptor | ≈28 days | Information not found | Ventricular dysfunction and congestive heart failure | [1] |
| Immunotherapy | | | | | |
| Atezolizumab | Humanized IgG monoclonal antibody that prevents interaction of PD-L1 and PD-1 | 27 days | Information not found | Fatigue, decreased appetite, nausea, urinary tract infection, pyrexia and constipation | [1] |



## 2. Materials and Methods

### 2.1. Literature Search

The search of literature for this review was made on the pubmed.gov database with the search terms: "metastatic breast cancer" OR "breast cancer" OR "triple-negative breast cancer", "epidemiology", "treatment" OR "therapy", "nanoparticles" OR "nanomedicine", "paclitaxel", "nab-paclitaxel" OR "nanoparticle albumin-stabilized paclitaxel". The investigation of the clinical trials was made on the clinicaltrials.gov database, with the search terms: "breast metastatic cancer" OR "nanoparticles".

### 2.2. Inclusion/Exclusion Criteria

This review included all the existing clinical trials that used nanoparticles in the treatment of metastatic breast cancer in humans. The selection of a clinical trial required the treatment of at least one group with nanoparticles, and the evaluation of its overall survival/outcome. Studies that used nanoparticles as a diagnostic imaging tool were not considered. In addition, clinical trials that included men in their participants were rejected. Since the purpose of this review is to gather information from recent reports, trials performed before 2010 were not included. No other exclusion criteria were applied.

### 2.3. Quality of Methods Assessment

The validity of the used clinical trial was assessed using JADAD score. This score calculates a total score for each trial based on randomization, blinding and account of all patients. For randomization, 1 point can be given if this randomization is mentioned, and another point can be given if the method of randomization is appropriate. For blinding, 1 point can be given if blinding is mentioned, another point can be given if the blinding method is appropriate. If the fate of all patients is known, 1 point is given in an account of all patients [17].

## 3. Results

### 3.1. Study Selection

In the search for this review, 75 clinical trials were found from the database on clinicaltrials.gov. After rejecting articles based on their title, abstract, and on their status, 30 articles remained. Articles preceding 2010 were rejected and one article was eliminated for having male patients. After applying these exclusion parameters, a total of 11 studies remained (Figure 2).

### 3.2. Study Characteristics

The clinical trials used in this review were mostly performed in the United States. One study was executed in the United States and Puerto Rico, and another was accomplished in Japan, Korea, and Taiwan. All patients were diagnosed with breast cancer, through methods of histology, imaging, or tumor markers; the majority of the patients had invasive/metastatic breast cancer and were previously treated with chemotherapy. The sample size of the studies had a range of 30–427 participants. Participant's ages ranged between 20 and 80 years; however, the age range was not mentioned in two studies. Mortality was also assessed, with a range of 5–31%; five studies did not mention the mortality rate and two studies only mentioned the number of dead patients, not the mortality rate. The follow-up period varied between 12 months and 8 years, and in two studies the length of this period was not mentioned (Table 2) (Figure 3). The overall median survival ranged between 39.7 weeks and 36.2 months; seven studies did not analyze overall median survival and instead evaluated parameters such as pathological complete response (pCR), progression-free survival (PFS), and overall clinical response. Adverse effects varied among the trials, but neutropenia, fatigue, diarrhea, and anemia were the most common drug side effects (Table 3).

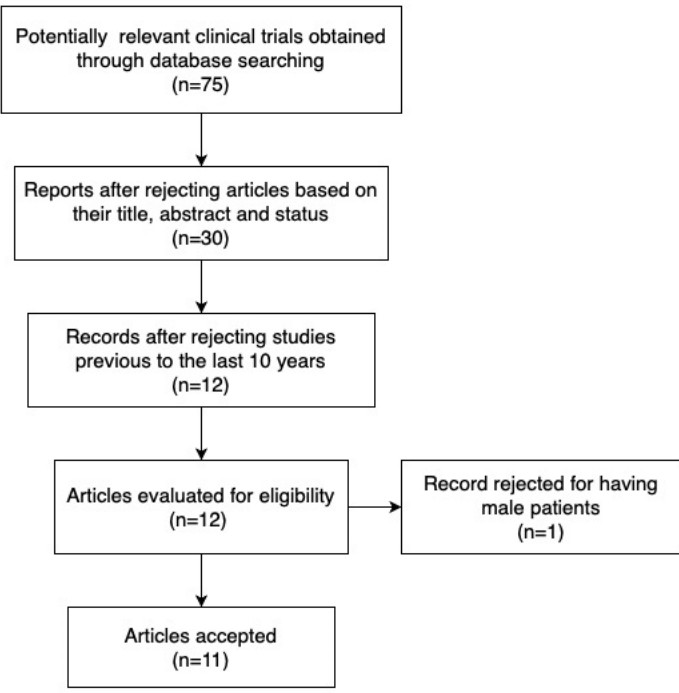

**Figure 2.** Flow chart demonstrating the selection method of clinical trials for this review.

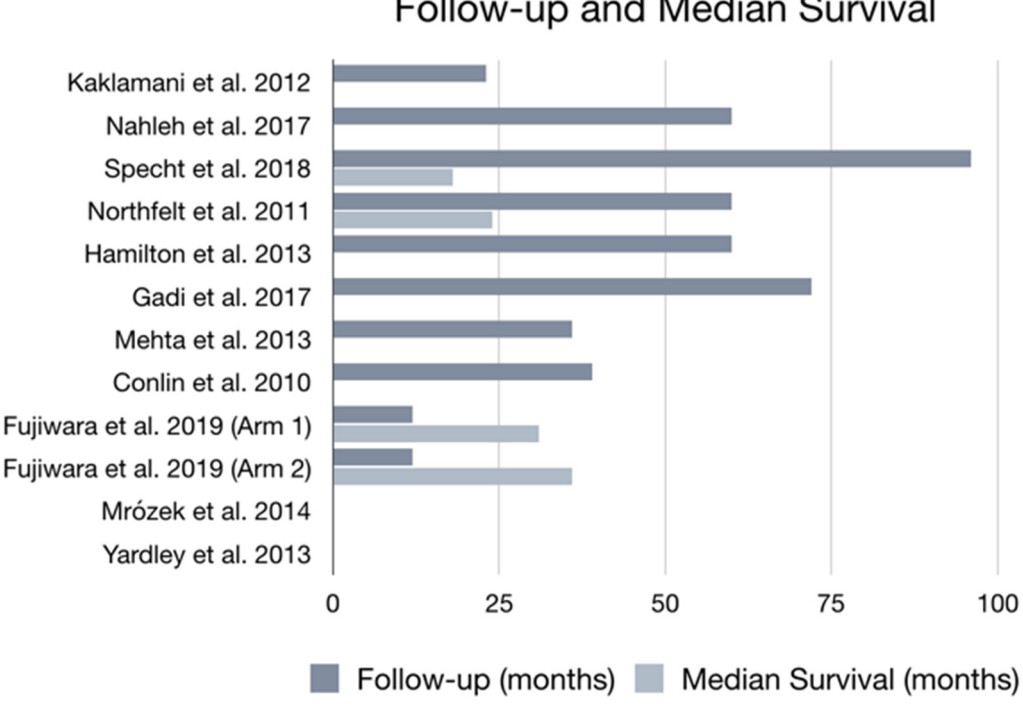

**Figure 3.** Follow-up and median survival of the reviewed clinical trials, in months.

**Table 2.** Characteristics of included clinical trials.

| Country/Region | Sample Size | Age Range (Years) | Previous Treatments | Follow-Up | Mortality | Assessment | Reference |
|---|---|---|---|---|---|---|---|
| United States | $n = 30$ | 27–70 | None | 22.6 months | 5 total deaths (16.67%) | Abraxane + lapatinib | [18] |
| United States and Puerto Rico | $n = 211$ | 22–75 | Not specified | 5 years | 31 total deaths 14 deaths in Arm 1 17 deaths in Arm 2/3 | Arm 1: nab-paclitaxel + bevacizumab followed by doxorubicin + cyclophosphamide + pegfilgrastim Arm 2: nab-paclitaxel followed by doxorubicin + cyclophosphamide + pegfilgrastim Arm 3: doxorubicin + cyclophosphamide + pegfilgrastim followed by nab-paclitaxel | [19] |
| United States | $n = 55$ | Not stated | Chemotherapy | 8 years | 3 total deaths (5.45%) | Nab-paclitaxel + bevacizumab, followed by bevacizumab + erlotinib hydrochloride | [20] |
| United States | $n = 60$ | 28–80 | Only one prior chemotherapeutic regimen | Not specified | 3 total deaths (5%) | Lapatinib + nab-paclitaxel | [21] |
| United States | $n = 33$ | 28–74 | Chemotherapy | Not specified | Not analyzed | Nab-paclitaxel + carboplatin and bevacizumab as neoadjuvant chemotherapy, followed by surgery and bevacizumab as adjuvant chemotherapy | [22] |
| United States | $n = 48$ | 27–77 | Chemotherapy, hormonal therapy, radiotherapy and immunologic therapy | 5 years | Not analyzed | Nab-paclitaxel + gemcitabine + bevacizumab | [23] |
| United States | $n = 41$ | 30–76 | Surgery, radiation and adjuvant chemotherapy | 5 years | Not analyzed | Abraxane + carboplatin + bevacizumab | [24] |
| United States | $n = 60$ | 29–69 | Surgery | 6 years | Not analyzed | Doxorubicin hydrochloride + cyclophosphamide + filgrastim followed by paclitaxel albumin-stabilized nanoparticle, patients with HER-2/NEU positive also receive trastuzumab | [25] |
| United States | $n = 43$ | Not stated | Not specified | 36 months | Not analyzed | Doxorubicin + cyclophosphamide + GM-CSF, followed by carboplatin + paclitaxel | [26] |
| United States | $n = 32$ | 29–76 | Surgery, radiation and chemotherapy | 39 months | 10 total deaths (31%) | Abraxane + carboplatin + trastuzumab | [27] |
| Japan, Korea and Taiwan | $n= 427$ | 20–74 | Chemotherapy | 12 months | 4 total deaths 3 in the paclitaxel group 1 in the treatment group | Arm 1: NK 105 Arm 2: Paclitaxel | [28] |

**Table 3.** Summary of results associating nanoparticles with breast cancer. Ten clinical trials used nab-paclitaxel in the treatment of breast cancer, one phase I study and nine phase II studies. One phase III clinical trial used a paclitaxel-incorporating micellar nanoparticle.

| Authors | Study Design | Nanoparticle Formulation | Selection Criteria | Main Objective | Participants | Overall Median Survival/Outcome | Adverse Reactions | Conclusion | Reference |
|---|---|---|---|---|---|---|---|---|---|
| Kaklamani et al. | Phase I Single Group Assignment | Abraxane (nanoparticle albumin-stabilized paclitaxel) | Stage I, II and III invasive breast cancer | Determining the efficacy of abraxane + lapatinib as neoadjuvant therapy in patients with stage I, II and III breast cancer. | n = 30. Age 27–70. | Not specified. pCR was obtained in 5 (17.9%) patients. | Diarrhea, neuropathy, fatigue, rash, bone pain, anemia, pruritus, fever, mucositis and vomiting | In general, the combination was well tolerated with minimal grade 3 toxicity and showed good efficacy. | [18] |
| Nahleh et al. | Phase II Randomized Clinical Trial | Nab-paclitaxel (nanoparticle albumin-stabilized paclitaxel) | Inflammatory and locally advanced HER2-/NEU negative breast cancer | Compare nab-paclitaxel, doxorubicin, cyclophosphamide and pegfilgrastim given with or without bevacizumab in the treatment of HER2-/NEU negative breast cancer. | n = 211. Age 22–75 | Not analyzed. Arm 1: pCR was obtained in 35 patients (35.7%) Arm 2/3: pCR was obtained in 24 patients (21.2%) | Anemia, febrile neutropenia, fatigue, watering eyes, constipation, diarrhea, nausea and mucositis oral | - | [19] |
| Specht et al. | Phase II Single Group Assignment | Nab-paclitaxel (nanoparticle albumin-stabilized paclitaxel) | Metastatic breast cancer | Evaluating maintenance treatment with erlotinib and bevacizumab after nab-paclitaxel and bevacizumab in women with metastatic breast cancer. | n = 55 | 18.1 months (95% CI, 15.6–21.7) | Infection, neutropenia, fatigue and neuropathy | - | [20] |
| Yardley et al. | Phase II Single Group Assignment | Nab-paclitaxel (nanoparticle albumin-stabilized paclitaxel) | HER2-positive metastatic breast cancer | Evaluate the efficacy and safety of nab-paclitaxel + lapatinib in women with HER2-postitive metastatic breast cancer who had received no more than one prior chemotherapeutic regimen. | n = 60. Age 28–80. | Not analyzed. PFS of 39.7 weeks (95% CI, 34.1–63.9) | Dehydration, diarrhea, anemia, cellulitis, febrile neutropenia, hypokalemia and acute renal failure | There is a clinical benefit for treatment with lapatinib and nab-paclitaxel. Toxicity was manageable and predictable. | [21] |
| Mrozek et al. | Phase II Single Group Assignment | Nab-paclitaxel (nanoparticle albumin-stabilized paclitaxel) | Stage II or III HER2-negative breast cancer | Determining the efficacy and safety of adding bevacizumab to the treatment with nab-paclitaxel and carboplatin in women with stage II or III HER2-negative breast cancer. | n = 33. Age 28–74. | Not analyzed. pCR was obtained in 6 patients. | Leukopenia, anemia, thrombocytopenia and neutropenia | The endpoint for efficacy was not reached. Regimen might be effective in TNBC. | [22] |
| Northfelt et al. | Phase II Single Group Assignment | Nab-paclitaxel (nanoparticle albumin-stabilized paclitaxel) | Metastatic breast cancer | Studying efficacy of treatment with nab-paclitaxel + gemcitabine + bevacizumab in metastatic breast cancer. | n = 48. Age 27–77. | 24.4 months (95% CI, 18.2–29.3) | Neutropenia, leukopenia, thrombocytopenia, anemia, dyspnea, diarrhea, nausea and nasal hemorrhage | The combination of nab-paclitaxel + gemcitabine + bevacizumab met the endpoint of 6 months PFS > 60%. Toxicity was manageable. | [23] |
| Hamilton et al. | Phase II Single Group Assignment | Abraxane (nanoparticle albumin-stabilized paclitaxel) | Triple-negative metastatic breast cancer (TNMBC) | Evaluate de efficacy of treatment with abraxane+ carboplatin + bevacizumab in TNMBC. | n = 41. Age 30–76. | Not specified. PFS of 9.2 months (95% CI, 7.8–25.1) | Neutropenia, fatigue, constipation, neuropathy, anemia, thrombocytopenia, alopecia and anorexia | The combination abraxane + carboplatin + bevacizumab is an active and tolerable regimen for first line TNMBC treatment. | [24] |
| Gadi et al. | Phase II Single Group Assignment | Nanoparticle albumin-stabilized paclitaxel | Breast cancer | Studying the efficacy of doxorubicin hydrochloride + cyclophosphamide + filgrastim, followed by nanoparticle albumin-stabilized paclitaxel with or without trastuzumab in patients with breast cancer previously treated with surgery. | n = 60. Age 29–69. | 59 surviving patients in 2 years 53 surviving patients in 6 years | Febrile neutropenia, fever, gastrointestinal disorders, dehydration and respiratory disorders | - | [25] |
| Mehta et al. | Phase II Single Group Assignment | Nab-paclitaxel (nanoparticle albumin-stabilized paclitaxel) | Breast cancer with 2cm and/or lymph node positive | Measuring the efficacy of treatment with doxorubicin + cyclophosphamide with GM-CSF, followed by carboplatin + nab-paclitaxel in breast cancer with 2 cm and/or lymph node positive. | n = 43 | Not specified. Overall clinical response was obtained in 43 patients. | Cardiovascular disease and neutropenic fever | - | [26] |
| Conlin et al. | Phase II Single Group Assignment | Abraxane (nanoparticle albumin-stabilized paclitaxel) | HER2-positive metastatic breast cancer | Evaluating the efficacy and safety of abraxane + carboplatin + trastuzumab in the treatment of HER2-positive metastatic breast cancer. | n = 32. Age 29–76. | NA; 81.3% of patients achieved t al response (95% CI, 67.7–94.8) | Neutropenia, anemia, nausea, thrombocytopenia, vomiting, diarrhea, constipation, fatigue, neuropathy and alopecia | The therapeutic regimen of abraxane + carboplatin + transtuzumab has high efficacy in HER2-overexpressing metastatic breast cancer, highlighting the advantage of a weekly taxane. | [27] |
| Fujiwara et al. | Phase III Randomized Clinical Trial | NK 105 (paclitaxel-incorporating micellar nanoparticle) | Metastatic or recurrent adenocarcinoma of the breast | Verify the non-inferiority of NK105 to paclitaxel in the treatment of metastatic or recurrent adenocarcinoma of the breast. | n = 427. Age 20–74. | Arm 1: 31.2 months (95% CI, 27.1–39.3) Arm 2: 36.2 months (95% CI, 30.3–NA) | Neutropenia, leukopenia, alopecia, neuropathy, rash, nausea, nasopharyngitis, diarrhea, fatigue, stomatitis, nail discoloration, myalgia and dysgeusia | Non inferiority of NK105 to paclitaxel was not demonstrated. Neuropathy profile was favorable. | [28] |

### 3.3. Synthesis of Study Results

Eleven clinical trials were assessed for survival and toxicity; ten of these clinical trials used a nanoparticle albumin-stabilized paclitaxel, and one used a micelle nanoparticle. The first ten clinical trials included a phase I study and nine phase II studies, and the micelle nanoparticle clinical trial was a phase III study.

### 3.4. Nanoparticle Albumin-Stabilized Paclitaxel

The first phase I clinical trial was performed by Kaklamani et al. and it determined the efficacy of abraxane (the first nab-paclitaxel drug approved by the US Food and Drug Administration) plus lapatinib as neoadjuvant therapy in 30 patients with stage I, II, and III breast cancer. This combination was feasible, with very few grade 3 toxicities such as rash, diarrhea, and fatigue and had a pCR of 17.9%. Though this trial had a small sample size, its results were comparable to other trials using this combination. The skin reactions in this study were similar to those previously described, and the authors emphasized the need for supportive dermatologic care for patients receiving lapatinib plus taxanes [18]. The first phase II trial was performed by Nahleh et al., and it aimed to compare nab-paclitaxel plus doxorubicin plus cyclophosphamide plus pegfilgrastim given with or without bevacizumab in the treatment of 211 women with HER2−/NEU negative breast cancer. pCR was obtained in 35.7% of the patients treated with bevacizumab and in 21.2% of the patients treated without bevacizumab. Adverse reactions such as anemia, febrile neutropenia, nausea, and vomiting were reported in both groups [19]. Another phase II trial was performed by Specht et al. and aimed to evaluate the efficacy of maintenance treatment with erlotinib plus bevacizumab after nab-paclitaxel plus bevacizumab in 55 women with metastatic breast cancer. Overall median survival was 18.1 months (95% CI, 15.6–21.7) with 5.45% mortality. Infection, neutropenia fatigue and neuropathy were the most common adverse effects [20]. Yardley et al. also developed a phase II trial to evaluate the efficacy and safety of treatment with abraxane and lapatinib in 60 women with HER2-positive metastatic breast cancer who had received no more than one prior chemotherapeutic regimen. In this study, a clinical benefit was witnessed for treatment with lapatinib and paclitaxel. A PFS of 39.7 (95% CI, 34.1–63.9) weeks shows clinically meaningful activity, with results consistent with other trials that assessed this combination therapy [29,30]. Adverse events had a toxicity grade of 3 or less, and most resolved without sequelae. Non-hematologic toxicities were consistent with those reported in another clinical trial [29]. This study established a dose regimen of nab-paclitaxel (100 mg/m$^2$ IV on day 1, 8, and 15 every 28 days) in combination with lapatinib (1000 mg orally once daily on a continuous basis) in a 4-week cycle with favorable efficacy and no additional safety indications [21]. Another phase II report was performed by Mrózek et al. where the efficacy and safety of adding bevacizumab to nab-paclitaxel plus carboplatin was determined in 33 women with stage II or III HER2-negative breast cancer. This study did not meet the expected pCR, the primary endpoint for efficacy. Still, for patients with TNBC, a longer course of neoadjuvant might increase the rate of pCR. This trial showed a higher than anticipated incidence of myelosuppression and the rate of complications was similar to that reported in other clinical trials that use bevacizumab in neoadjuvant treatment. This treatment should be further investigated [22]. Northfelt et al. reported a phase II trial to study the efficacy of treatment with nab-paclitaxel plus gemcitabine plus bevacizumab in 48 women with metastatic breast cancer. This study met its endpoint, with 6 months PFS >60%, and had an overall survival of 24.4 months (95% CI, 18.2–29.3). Therefore, this therapeutic combination was concluded to be effective in patients with metastatic breast cancer. Toxicities were manageable, permitting long-duration therapy. The most common adverse effects were neutropenia, fatigue, thrombocytopenia, anemia, leukopenia, and dyspnea. This regimen should be further evaluated in regimens of chemotherapy plus targeted agents in metastatic breast cancer [23]. Another phase II study was performed by Hamilton et al. to evaluate the efficacy of treatment with abraxane plus carboplatin plus bevacizumab in 41 women with TNMBC. This therapeutic regimen showed activity and tolerability

for first-line treatment of TNMBC, with a PFS of 9.2 months, similar to other trials that used this regimen in other subtypes of breast cancer. Rates of febrile neutropenia and neuropathy were comparable to those undergoing similar treatments. This study is worthy of further evaluation [24]. Gadi et al. developed a phase II trial aimed to determine the efficacy of doxorubicin hydrochloride plus cyclophosphamide plus filgrastrim, followed by nanoparticle-albumin stabilized paclitaxel with or without trastuzumab in 60 patients with breast cancer previously treated with surgery. There were 59 surviving patients in 2 years and 53 surviving patients in 6 years. Adverse effects such as febrile neutropenia, fever, gastrointestinal disorders, dehydration, and respiratory disorders occurred during the treatment [25]. Mehta et al. measured the efficacy of a treatment regimen of doxorubicin plus cyclophosphamide plus granulocyte-macrophage colony-stimulating factor (GM-CSF), followed by carboplatin plus nab-paclitaxel in 43 women with 2 cm and +/or lymph node-positive breast cancer in a phase II trial. Overall clinical response was obtained in 43 patients in the time frame of 3 years. Cardiovascular disease and neutropenic fever were the most common adverse effects reported [26]. Conlin et al. developed a phase II trial aiming to evaluate the efficacy and safety of abraxane in combination with carboplatin and trastuzumab in the treatment of 32 women with HER2-positive metastatic breast cancer. This trial showed high efficacy in HER2-overexpressing metastatic breast cancer, with a total pathological response achieved in 81.3% of the patients and emphasized the advantage of a weekly taxane. This regimen was very well tolerated, with neutropenia and leukopenia being the most common toxicities witnessed [27].

### 3.5. Micelle Nanoparticles

Only one clinical trial was found to use micelle nanoparticles. NK105 is a paclitaxel-incorporating polymeric micellar nanoparticle with a notable EPR effect [28]. Studies in rodent models have shown that the extended circulation of NK105 allows a higher accumulation in the tumor than free paclitaxel. However, doubts remain about the accumulation of the anticancer drug leading to exercising its function in vivo [31].

A phase III study was performed by Fujiwara et al. and aimed to verify the non-inferiority of NK105 to paclitaxel in the treatment of 427 women with metastatic or recurrent adenocarcinoma of the breast. Overall survival of patients treated with NK105 in comparison to the control group was 31.2 months (95% CI, 27.1–39.3) vs. 36.2 (95% CI, 30.0–NA). The expected superior efficacy of paclitaxel, due to the EPR effect reported in a non-clinical study, was not achieved. Therefore, this study did not demonstrate the superiority of NK105 compared to paclitaxel. Hematological toxicity was similar in both groups, but the profile for non-hematological toxicity was better in the NK105 group. Only a few grade 3 infections occurred in both groups, and neutropenia was reported in only one patient in each group. The efficacy of NK105 should be reassessed in future studies [28].

### 3.6. Quality of Used Clinical Trials

The quality of the clinical trials was measured on Table 4 using the JADAD score. Apart from the trials performed by Nahleh et al. and Fujiwara et al., which were randomized clinical trials, all clinical trials had a single group assessment intervention model. Therefore, there was no randomization or blinding. We consider that as a limitation and a weakness in all of the studies.

**Table 4.** Quality assessment of the reviewed studies using JADAD score.

| Author/Principal Investigator and Year | Randomization | Blinding | An Account of All Patients | Total Score |
|---|---|---|---|---|
| Kaklamani et al., 2012 | 0 | 0 | 1 | 1 |
| Nahleh et al., 2017 | 1 | 0 | NA | 1 |
| Specht et al., 2018 | 0 | 0 | NA | 0 |
| Yardley et al., 2013 | 0 | 0 | 1 | 1 |
| Mrózek et al., 2014 | 0 | 0 | 1 | 1 |
| Northfelt et al., 2011 | 0 | 0 | NA | 0 |
| Hamilton et al., 2013 | 0 | 0 | 1 | 1 |
| Gadi et al., 2017 | 0 | 0 | 1 | 1 |
| Mehta et al., 2013 | 0 | 0 | 1 | 1 |
| Conlin et al., 2010 | 0 | 0 | 1 | 1 |
| Fujiwara et al., 2019 | 2 | 0 | 1 | 3 |

## 4. Discussion

Breast cancer remains a major cause of mortality in women around the globe [2]. Breast cancer has been treated with surgery, chemotherapy, radiotherapy, hormonal therapy, and immunotherapy according to its subtype, grade, and stage. Nevertheless, to date, it is still difficult to completely treat this disease due to its heterogeneity. The use of nanomedicine in the treatment of cancer revolutionized the field and breast cancer currently has the highest number of clinical trials using nanoparticles [32]. The technology of encapsulating chemotherapeutic drugs increases their bioavailability and decreases adverse effects, therefore increasing efficiency and decreasing toxicity [33]. This review summarizes the findings of 11 clinical trials that use nanoparticles in the treatment of breast cancer, performed since 2010.

In the study reported by Mrózek et al., weekly treatment with nab-paclitaxel plus bevacizumab could not prove the higher efficacy. The reason for these unexpected results could be the lack of a control arm and the relatively small sample size. Although the primary endpoint for efficacy was not met, patients with TNBC had a pCR rate of 50%, suggesting that this regimen could be effective in the treatment of patients with TNBC. Therefore, more studies should be performed to further evaluate this information [22]. In the study performed by Fujiwara et al., the clinical benefit of NK105 compared to paclitaxel was not proven. NK105 was expected to be a more efficient treatment due to the EPR effect that had been reported in a non-clinical study. However, the reason for not reaching the primary endpoint of this study could be the lower dose intensity of NK105. Nonetheless, peripheral sensory neuropathy profile was favorable with NK105 treatment in comparison to paclitaxel [28].

Even though nanoparticles are not frequently used in clinical treatment yet, there is a large amount of evidence that proves its benefit in terms of increased drug half-life, better solubility, improved drug accumulation on the tumor site and reduction in adverse effects [34]. The trial performed by Kaklamani et al. showed a pCR of 17.9%, a result similar to a previously reported clinical trial that used lapatinib in combination with weekly paclitaxel in women with inflammatory breast cancer, however, it showed a decrease in grade 3 diarrhea [20,35]. The use of abraxane has also been reported to improve overall response and survival, as well as to decrease hypersensitivity reactions, reducing the use of corticosteroids in advance [36]. The trial performed by Yardley et al. reported a lower overall response rate in comparison to similar studies. However, the primary endpoint PFS was 39.7 weeks, a result which is consistent with two similar studies that showed PFS of 41.7 weeks [29] and 47.9 weeks [30]. The trial performed by Northfelt et al. met its prespecified endpoint of 6 months PFS > 60% and its toxicities were manageable [23]. The trial performed by Halmilton showed PFS of 9.2 months in comparison to a similar study with PFS of 6.1 months [37,38], and also described a favorable rate of febrile neutropenia (3%) and neuropathy (6% grade 3, no grade 4) in comparison to historical control treatments [24]. In accordance with these results, multiple studies on

breast cancer with abraxane have shown longer time to tumor progression and less grade 4 neutropenia incidence [38,39]. The trial performed by Conlin reported PFS of 16.6 months in comparison to a previous study with PFS of 14.7 months for weekly paclitaxel [40], and also a 62% rate of patients with grade 1 or greater peripheral neuropathy (vs. 73% in the previous trial). Overall, five out of the eleven reviewed clinical trials showed a clinical benefit in the use of nanoparticle albumin-stabilized paclitaxel for the treatment of breast cancer.

### 4.1. Limitations

There was heterogeneity among the trials used in this review, in terms of the type of breast cancer, dosage, and treatment regimen. These differences can be responsible for dissimilar results of efficiency and toxicity among trials. As mentioned before, there are currently multiple types of nanoparticle in development, with a portion of them not mentioned in this review. An additional limitation was the small sample size in some studies, though the authors justified most of them.

### 4.2. Future Research

Throughout our analyses, we could detect a lack of standardization in the treatment regimen, which could lead to different results of efficiency and toxicity. Although efficient, this type of treatment is still not commonly used in a clinical context. Nevertheless, with the increasing scientific knowledge on the field regarding the safety and efficacy of these systems, the development of novel types of nanoparticles with improved characteristics, and the design of clinical trials with more standardized treatment regimens, an increase in the number of clinical trials with nanoparticles is predictable. Consequently, in the near future, it is expected that nanomedicine may have a more significant contribution in clinical routines for the treatment of metastatic breast cancer, and some formulations may finally reach phase IV of the clinical trials and therefore constitute a tool for the adjuvant and neoadjuvant treatment of this systemic and poor prognosis disease.

### 5. Conclusions

Multiple clinical trials that use nanoparticles in the treatment of metastatic breast cancer have been performed; 10 out of the 11 reviewed clinical trials used nanoparticle albumin-stabilized paclitaxel. Five out of the eleven reviewed clinical trials confirmed the benefit of using this emerging treatment, in combination with chemotherapy and targeted therapy, whilst reducing toxicity. The authors of the clinical trials that did not show benefit in the use of nanoparticles advised further evaluation with a higher dosage and/or superior sample size to be tested.

**Author Contributions:** Conceptualization, M.P.; methodology, M.P.; investigation, R.M. and M.P.; writing—original draft preparation, R.M.; writing—review and editing, R.M., A.G., M.P. and S.R.; supervision, M.P. All authors have read and agreed to the published version of the manuscript.

**Funding:** This research received no external funding.

**Conflicts of Interest:** The authors declare no conflict of interest.

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
