# Peer review of "Nanomedicine Interventions in Clinical Trials for the Treatment of Metastatic Breast Cancer"

_applsci, doi:10.3390/app11041624_

Round 1

Reviewer 1 Report

The review entitled Emerging Nanomedicine Interventions in Metastatic Breast Cancer submitted by Rita Moreira et al was sought, according to the authors, “to gather information and analyze recent clinical trials evaluating the therapeutic effects of nano-particles in the treatment of metastatic breast cancer”. However, the authors substantially fail to transpose the interesting, promising title into a corresponding comprehensive review. In my opinion, the manuscript should be redesigned and rewritten in order to become acceptable for publication.

Some comments/observations regarding the shortcomings that argue my decision:

  1. Despite the wide horizon of the title, the review is essentially dedicated to nanoparticle-stabilized paclitaxel only; a discussion about a database with 10 albumin- and 1 micelle-stabilized paclitaxel could not be considered a review about “Emerging Nanomedicine Interventions”, but just a mini-review on emerging albumin-stabilized paclitaxel formulations in clinical trials;
  2. Too much emphasis goes on well-known drugs used in metastatic breast cancer and disproportionately low info regards nano-formulations;
  3. Discussions and approaches are relatively shallow and encompass limited points of view. The authors are encouraged to read and compare their work with other reviews in the area; few arbitrary examples could be - doi: 10.1016/j.biopha.2017.09.059; doi: 10.1002/btm2.10143; doi: 10.2217/nnm-2018-0120; doi: 10.1021/acs.accounts.9b00228; doi: 10.1177/1559325820936161;
  4. A review of “Emerging Nanomedicine Interventions” must include and discuss other types of nanoparticles, with other drugs and different nano-formulations, thus giving to a reader an image as complete as possible. For example, how about other active nano-therapeutics like RNA, polymer and metal-based ones? If recent clinical trials were not found for various reasons, pre-clinical and/or emerging perspectives should be pointed out at least.

Taken into account these considerations, as mentioned above, this manuscript is not acceptable for publication in Applied Sciences Journal unless the authors decide to substantially revise the content of their submission.  

Author Response

Referee 1:

 “The review entitled Emerging Nanomedicine Interventions in Metastatic Breast Cancer submitted by Rita Moreira et al was sought, according to the authors, “to gather information and analyze recent clinical trials evaluating the therapeutic effects of nano-particles in the treatment of metastatic breast cancer”. However, the authors substantially fail to transpose the interesting, promising title into a corresponding comprehensive review. In my opinion, the manuscript should be redesigned and rewritten in order to become acceptable for publication”.

Comment 1

“Some comments/observations regarding the shortcomings that argue my decision: Despite the wide horizon of the title, the review is essentially dedicated to nanoparticle-stabilized paclitaxel only; a discussion about a database with 10 albumin- and 1 micelle-stabilized paclitaxel could not be considered a review about “Emerging Nanomedicine Interventions”, but just a mini-review on emerging albumin-stabilized paclitaxel formulations in clinical trials”;

We understand the reviewer’s concerns. We have recently published another review paper where our main focus was the use of nanocarriers for the treatment of breast cancer in a pre-clinical perspective. In this paper, however, we provide a deeper perspective of the nanomedicine but in the clinical trials for the treatment of metastatic breast cancer, highlighting the potential candidates to be useful in the clinical practice to be useful in this disease. Since we are in agreement with the referee the title of the manuscript was changed to be more accurate and also the manuscript was rewritten to highlight that the focus of this manuscript was in the clinical trials.

Comment 2

“Too much emphasis goes on well-known drugs used in metastatic breast cancer and disproportionately low info regards nano-formulations”;

We understand the reviewer’s concerns. As we stated before in this manuscript we inserted all the clinical trials that exist in this field and the idea of this review was to highlight the nanomedicine interventions already in the clinical trials and not the pre-clinical studies, which is already a theme widely covered in the literature, including by cited reviews and mentioned in this manuscript. In addition, the introductory section is focused on the already established drugs used in clinical practise as this review aims to provide the reader with a clinical perspective of the treatment of this disease.  

Comment 3

“Discussions and approaches are relatively shallow and encompass limited points of view. The authors are encouraged to read and compare their work with other reviews in the area; few arbitrary examples could be - doi: 10.1016/j.biopha.2017.09.059; doi: 10.1002/btm2.10143; doi: 10.2217/nnm-2018-0120; doi: 10.1021/acs.accounts.9b00228; doi: 10.1177/1559325820936161”.

We are grateful for the reviewer suggestions. We introduced new information in the manuscript, and we have compared with other reviews, including among other the valuable suggestions of the referee.

Comment 4

“A review of “Emerging Nanomedicine Interventions” must include and discuss other types of nanoparticles, with other drugs and different nano-formulations, thus giving to a reader an image as complete as possible. For example, how about other active nano-therapeutics like RNA, polymer and metal-based ones? If recent clinical trials were not found for various reasons, pre-clinical and/or emerging perspectives should be pointed out at least”.

We understand the referee concern. However, and according our search the nanoformulations developed and in the clinical trials in the last 10 years were the NPs described in this review. Thus, the other types of NPs with only preclinical results are not the focus of this review as we previously stated. Taking in consideration the recommendation, new information was inserted in the section of methods about this topic.

Reviewer 2 Report

Dear Editor,

I carefully read the manuscript entitled “Emerging Nanomedicine Interventions in Metastatic Breast Cancer” by Rita Moreira et al. that deal with dissertation on  clinical trials that use nanoparticles in the treatment of metastatic breast cancer.

Overall, there are some misleading sentence in the manuscript and the research method appear too restrictive, thus the idea is that this particular manuscript can be easily interpretated as a review witch speak about abraxane clinical trials. For instance, why the authors have not used the word theranostics in the research criteria? Another question is that the authors have claimed that there are not precision therapies proposed for triple negative breast cancer, but this is not true. There are many immunological approaches proposed for precision approaches which deserve attention.

By the way, please find in the follows some suggestion to improve the manuscript:

  • Please enclose in Table 1 other drugs commonly used in the treatment of breast cancer, such as irinotecan or somatostatin analogues, and try to classify them on the basis of the stage and also as combination therapy;
  • The title should be changed considering that this particular review is focused on clinical trials;
  • The authors should add clinical trials on theranostic nanoparticles such as SPIONs, thus involving magnetic nanoparticles that stick to breast cancer cells.

Author Response

Referee 2:

Comment 1

“I carefully read the manuscript entitled “Emerging Nanomedicine Interventions in Metastatic Breast Cancer” by Rita Moreira et al. that deal with dissertation on clinical trials that use nanoparticles in the treatment of metastatic breast cancer. Overall, there are some misleading sentence in the manuscript and the research method appear too restrictive, thus the idea is that this particular manuscript can be easily interpretated as a review which speak about abraxane clinical trials. For instance, why the authors have not used the word theranostics in the research criteria? Another question is that the authors have claimed that there are not precision therapies proposed for triple negative breast cancer, but this is not true. There are many immunological approaches proposed for precision approaches which deserve attention”.

We understand the reviewer’s concerns. In fact, there are several review papers where the main focus was the use of nanoparticles for the treatment of breast cancer with the preclinical results highlighted. In this paper, however, we provide a deeper perspective of the nanomedicine but in the clinical trials for the treatment of metastatic breast cancer, highlighting the potential candidates to be useful in the clinical practice and the particular characteristic of the clinical trials. Since we are in agreement with the referee the title of the manuscript was changed to be more accurate and also the manuscript was rewritten. The use of theranostics in clinical trials of metastatic breast cancer was not mentioned since no clinical trial was found regarding this subject and, in addition, this manuscript is focused on the treatment. For that reason, the title was rewritten to be more accurate with the objective of this review. Regarding the last comment we completely agree with the referee and we thank the referee for the correction, being the sentence that claimed that immunotherapy is not a precision therapy eliminated.

Comment 2

“By the way, please find in the follows some suggestion to improve the manuscript: Please enclose in Table 1 other drugs commonly used in the treatment of breast cancer, such as irinotecan or somatostatin analogues, and try to classify them on the basis of the stage and also as combination therapy”;

We completely agree with the referee and, therefore, we have inserted other drugs in Table 1 used in the treatment of breast cancer, including irinotecan and ocreotide.

Comment 3

“The title should be changed considering that this particular review is focused on clinical trials”;

We completely agree with the referee and the title was changed to include the clinical trials main focus.

Comment 4

“The authors should add clinical trials on theranostic nanoparticles such as SPIONs, thus involving magnetic nanoparticles that stick to breast cancer cells”.

According to the above-mentioned the use of SPIONs in clinical trials of metastatic breast cancer was not mentioned since no clinical trial was found regarding this subject for the metastatic breast cancer. However, and since the theranostic using SPIONs is an important and potential tool we inserted information in the introduction about the preclinical results of SPIONs in the manuscript.

Round 2

Reviewer 2 Report

I only suggest the authors to deeply check literature data. For instance, here there is a Multicenter clinical trial on sentinel lymph node biopsy using superparamagnetic iron oxide nanoparticles and a novel handheld magnetic probe (J surg onc 2019 Dec;120(8):1391-1396) that should be cited in this particular review. 

Author Response

The authors acknowledge the suggestion of the referee. We double check the literature and we did not found any new clinical trial for the treatment of metastatic breast cancer. Regarding the suggestion, the clinical trial is focused on the diagnosis and not in the treatment and moreover is related with breast cancer and not metastatic breast cancer. However, since it is an important reference we add the citation in this manuscript.
